

**Quantification of uncertainty in rapid estimation of earthquake fatalities**
**based on scenario analysis**
Xiaoxue Zhang[1,2], Hanping Zhao[1,2], Fangping Wang[1,2], Zezheng Yan[1,2], Sida Cai[1,2] & Xiaowen Mei[1,2]
1.Key Laboratory of Environmental Change and Natural Disaster, MOE, Faculty of Geographical Science,
Beijing Normal University, Beijing, China;
2.Academy of Disaster Reduction and Emergency Management, Ministry of Civil Affairs & Ministry of
Education, Faculty of Geographical Science, Beijing, China;
Correspondence to: Hanping Zhao (zhaohanping@bnu.edu.cn)
**Abstract:** The rapid estimation of earthquake fatalities using earthquake parameters is the core
basis for emergency response. However, there are numerous factors affecting earthquake
fatalities, and it is impossible to obtain an accurate estimation result. The key to solve this
problem is quantifying the uncertainty. In this paper, we proposed a new method to estimate
earthquake fatalities and quantify the uncertainty based on basic earthquake emergency scenarios.
The accuracy of the model is verified by earthquake that occurred during recent year. The
preliminary analysis and comparison results show that the model is more effective and reasonable
and can also provide a theoretical basis for post-earthquake emergency response.
**Keywords:** earthquake fatalities, rapid estimation, scenario analysis, uncertainty, information
diffusion
**1 Introduction**

20       The most important assessment after a destructive earthquake is the estimation of fatalities

(Samardjieva. 2002). However, a field investigation cannot be conducted quickly, often because
of road damage and communication interruption. (Kongar et al. 2015; Yuan and Wang. 2009).
Nevertheless, one can estimate earthquake fatalities in a few minutes using earthquake
parameters (such as magnitude, intensity and initial time) (Frolova, et al. 2011; Wald, et al. 2008).
In addition, it is essential to study the uncertainty of the estimation because there are various
uncontrollable factors in the process of estimation. In this sense, a preliminary estimation with
uncertainty analysis of earthquake fatalities using available earthquake parameters is a key path
in starting the emergency response.

29       At present, the methods for estimating earthquake fatalities mainly include analytical, semi-



analytical and empirical models (Federal Emergency Management Agency (FEMA), 2005).

However, the calculation of analytical and semi-analytical models are based on building damage

data, which are not suitable for rapid estimation (Li, et al. 2015; Weng, et al. 2009). During recent

years, the empirical model has been widely used in rapid estimation, which depends on statistical

analysis using historical loss data. The empirical model provides an important opportunity to

quickly and approximately assess the earthquake loss. Regarding the study of the empirical

model, Japanese researchers did so relatively early. Kawasumi (1951) proposed a measure to

estimate the danger and expectation of the maximum intensity of destructive earthquakes in Japan.

Similarly, Ohta et al. (1983) developed an empirical relationship for estimating the number of

casualties within the number of completely destroyed houses. A more recent attempt was based

on an analysis of strong global earthquakes during the twentieth century, which obtained a log-

linear relationship for fatalities as a function of magnitude and population density (Samardjieva.

2002). On the basis of Samardjieva's study, Badal et al. (2005) put forward a quantitative

earthquake fatality estimation model that considered the mortality rate. Similarly, Nichols and

Beavers (2003) studied the earthquake loss catalog of the twentieth century and established a

bounding function with the fatality count and magnitude. Chen et al. (2005) analyzed earthquake

cases on mainland China and developed an empirical equation based on the standard of

population density and the relationship between the seismic fatalities and the magnitude. Jaiswal

et al. (2009) established a mortality model based on population distribution according to rebuilt

earthquake case scenes and studied regional earthquake cases (Jaiswal et al. 2010). Generally

speaking, the current empirical model for fatality estimation is derived from available historical

data and relies on parameter regression analysis. Therefore, there are two problems with the

empirical model. First, it will ignore extreme events when there is lack of historical data. Second,

most models consider fewer factors and do not consider the influence between know factors and

possible unknown factors. It is quite essential to establish a new rapid estimation model of

earthquake fatalities that can avoid these problems.

The data or processes used in the empirical model contain considerable uncertainty, and the

uncertainty in these components is the source of inaccuracy or error in the estimation results

(Gardi et al. 2011; Gall et al. 2009; Wirtz et al. 2014). During recent years, the study of

uncertainty in the estimation of earthquake fatalities has mainly regarded the qualitative





description (Romão, 2016), and there is a relative lack of quantitative research. Qualitative
description is the most widely used method to describe the uncertainty in disaster estimation (Van
Asselt 2000). There are many linguistic uncertainties when describing the uncertainty in terms
of vagueness and context, which can result in an inaccurate qualitative description. The numerical
quantification of uncertainty is possible for emergency decision making when the information is
partial or not quantifiable during the process of estimation. It is imperative to construct a suitable
model to quantify the uncertainty in the estimation of earthquake fatalities.

67        In this paper, we present a new approach to estimate earthquake fatality expectations and

quantify the uncertainty in the estimation, which is expressed as a function of the mortality rate
and victims. The basic scenarios are constructed using the magnitude, the initial time and the
relationship between the epicentral intensity and the epicentral fortification intensity, and these
scenarios consider combinations of parameters. This study not only breaks the traditional
empirical model form but also quantifies the uncertainty in the estimation results.

## 2 Earthquake fatalities in mainland China

74        In general, historical earthquake fatality and exposure data provide a useful basis for future

earthquake fatality estimation. We collected destructive earthquake data from earthquakes that
occurred on mainland China from 1970 to 2017 as samples. The datasets mainly contain the
earthquake parameters (e.g., magnitude, epicentral intensity, epicentral fortification intensity and
initial time) and the disaster information (e.g., the number of fatalities and the number of victims);
the distribution of the samples is shown in Figure 1. The disaster information was derived from
EM-DAT (http://www.emdat.be/), and the earthquake parameters were obtained from PAGER
(https://www.pager.com/).





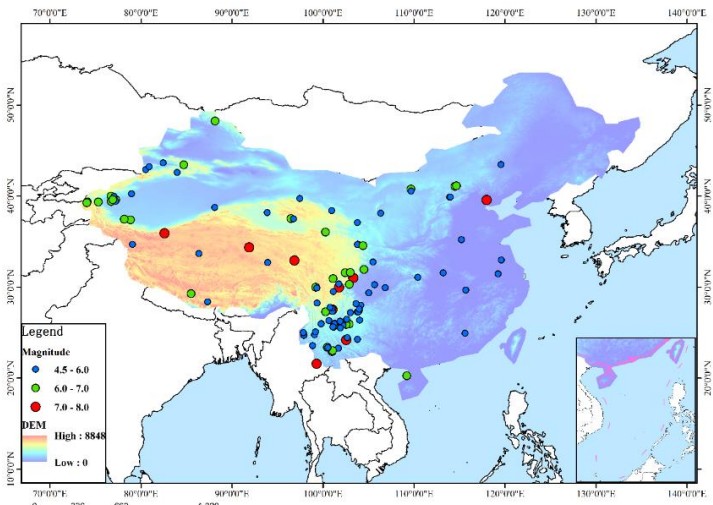


**Figure 1. Distribution of historical earthquakes on mainland China from 1970 to 2017**


## 3 Basic earthquake emergency scenarios


Scholars have discussed the factors that affect earthquake fatalities, which include
magnitude, intensity, initial time, population exposure, housing fragility, and individual factors
(Oike, 1991; Nichols, 2003). Moreover, scholars have considered as many factors as they can
when modelling. However, some errors remain in each model; thus, the relational expression
between the parameters and the number of fatalities is not suitable, or there are still some
temporarily non-measurable factors. Therefore, we hoped to identify the main influencing factors
via the analysis of historical data. Basic earthquake emergency scenarios were constructed based
on a combination of the main factors. A basic scenario combination can better express the
relationship between the parameters and earthquake fatalities. Then, information diffusion theory
was used to diffuse the sample data based on the basic scenarios considering the temporarily non-
measurable factors and the extreme event under each scenario.
We collected data on 219 destructive earthquakes that caused casualties in China from 1970
to 2017. Via qualitative analysis using the collected data, the main factors affecting earthquake
fatalities were acquired. There is an approximately linear relationship between the magnitude
and the number of fatalities (Figure 2). As the magnitude increases, the number of fatalities
increases. The relationship between the epicentral intensity and the number of fatalities is shown
in Figure 3; the epicentral intensity is mapped to the number of fatalities. The relationship



between the number of fatalities and the initial time is relatively vague, as shown in Figure 4.
However, it is evident that the maximum number of fatalities occurred during the period 21:00-
06:00. The initial time of the earthquake will influence the in-building ratio, the population
exposure and the speed of the escape reaction of indoor personnel (Chen 1993; Yang et al. 2007).
After analysis, it was found that there was no ideal correspondence between the collapse area
and the number of fatalities, as shown in Figure 5.

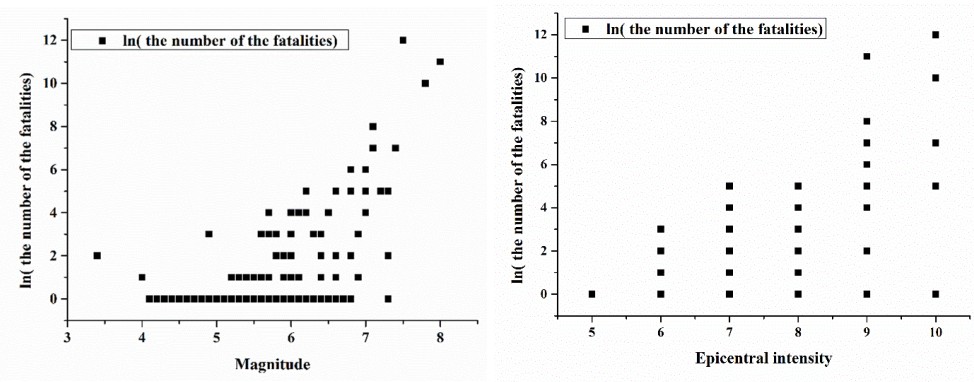

Figure 2. Relationship between the magnitude and the

number of fatalities

Figure 3. Relationship between the epicentral intensity

and the number of fatalities

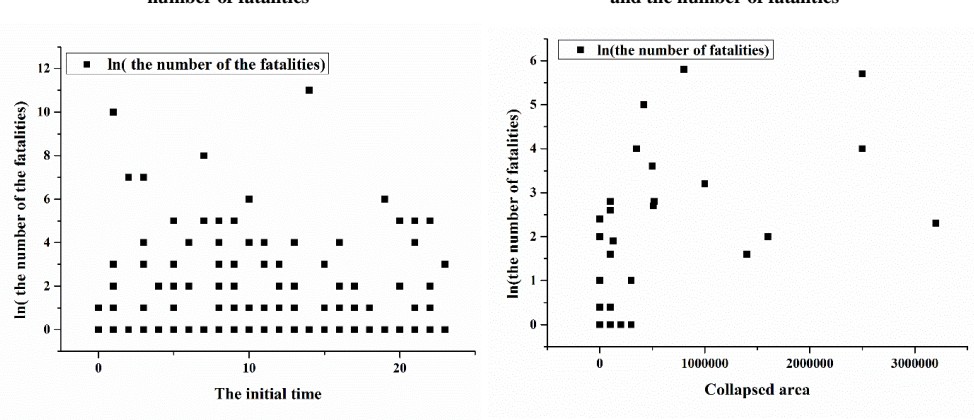

Figure 4. Relationship between the initial time and the

number of fatalities

Figure 5. Relationship between the collapsed area and

the number of fatalities

Based on the aforementioned analysis, the magnitude, epicentral intensity and initial time
were selected as the main parameters used to establish the basic earthquake emergency scenarios.
Magnitude can be expected to be the most essential factor in determining earthquake fatalities.
The magnitude was divided into three levels ($4.5 \leq M < 6$, $6 \leq M < 7$ and $7 \leq M \leq 8$ (M means

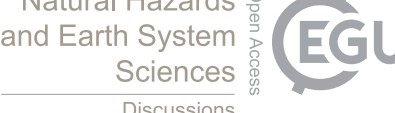



magnitude)) according to the principle of magnitude division in the earthquake emergency
programming of China (The National Earthquake Emergency Plan, 2012). On the basis of the
magnitude division, the relationship between the empricial intensity and the fortification intensity
was used to indirectly express the building damage information. The relationship between
magnitude (M) and epicentral intensity ($I_0$) is as follows : $M = 0.58I_0 + 1.5$ (GB/T17742). As
the fomula shows, when the magnitude is greater than 6, the empirical intensity is greater than
7.75. However, there are fewer historical earthquakes with a regional fortification intensity
greater than 8 in China. Therefore, the basic earthquake emergency scenarios do not consider the
scenario with an epicentral intensity less than the epicentral fortification intensity when the
magnitude is greater than 6. In addition, the initial time of the earthquake is an important factor
affecting staff reaction. During early morning or night, most of the population is sleeping in
residential buildings; thus, they cannot take protective measures. In contrast, during the day, most
of the population is at work. Thus, the initial time was devided into two periods: day (06:00-
20:59) and night (21:00-05:59). Finally, the basic earthquake emergency scenarios were
constructed based on a combination of the magnitude, intensity, and initial time of the earthquake
(Figure 6).

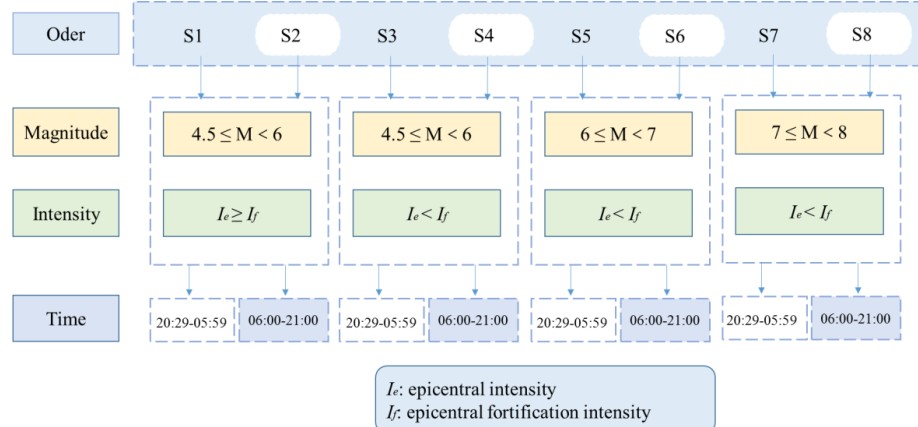


**Figure 6. Framework of basic earthquake emergency scenarios**
The objective of the rapid estimation model of earthquake fatalities based on scenario
analysis is to estimate the fatality expections and the uncertainty in the fatality interval. The
sample data were classified into each scenario based on the framework of the basic earthquake
emergency scenarios. Then, the classified samples were devided into two sets (Table 1). One set

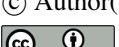


consisted of 80% of sample data, which were selected randomly selected from each scenario for
model construction. Another set was composed of the remaining 20% of the samples, which was
used to verify the accuracy of the model.

**Table 1. Data sample size and data usage**

| Sample size | Data usage |
|---|---|
| 175 (random selection of 80% of the samples under each scenario) | Model construction |
| 44 (random selection of 20% of the samples under each senario) | Verification |

## 4 Methodology
We needed a functional form describing the fatalities with the victim and morality rate.
After the earthquake, the China Earthquake Administration will rapidly publish information on
the earthquake, including the magnitude, the geographic coordinates of the epicentre, and the
source mechanism solution (Wang, et al. 2013). The intensity distribution is acquired by the
earhquake parameter information and the seismic intensity elliptical attenuation model (Wang,
et al. 2000; Wu, et al. 2010). The number of victims is calculated with the area of each intensity
and the population density. To derive an earthquake fatality rapid estimation function, one needs
to compile the mortality rate statistical analysis under each scenario using observations from past
earthquakes. The outline of the approach is as follows:

$$D = E(S_t) \times \sum_{I=5}^{I_{max}} k_I A_I P_I \qquad (1)$$

where D is the number of fatalities; $E(S_t)$ is the mortality rate expectation of scenario $S_t$;
$A_I$ is the affected area of the intensity $I$; $I_{max}$ is the maximum intensity for an earthquake; $P_I$ is
the population density of the intensity $I$, and parameter $k_I$ is the ratio of the population affected
by the earthquake, as determined from the damage degree table provided by the National Disaster
Reduction Center (Fan et al., 2008).
To obtain the mortality rate function beyond the framework of the basic earthquake
emergency scenarios, we needed to use the observed data of historical earthquakes to compile a
mortality rate expectation under each scenario. However, when dividing the samples into each



scenario, the sample size will be small, and it is difficult to obtain the relation equation using
traditional mathematical statistics. Therefore, the indirect approach of this study consisted of
information diffusion theory to obtain the mortality rate. First, the actual observed values for the
mortality rate under one scenario were set as matrix $X = \{x_1, x_2, ..., x_m\}$, where $x_i$ is the actual
observed values of an earthquake, and $m$ is the total number of earthquake events. At the same
time, the actual recorded mortality rate and historical extreme event (the earthquake event with
an extreme mortality rate) under one scenario were considered to build the domain $U =$
$\{u_1, u_2, u_3, ..., u_n\}$. Here, $u_j$ is the arbitrary discrete real value in the interval $[u_1, u_n]$, and $n$ is
the total number of discrete points. Then, the sample value $x_i$ was diffused to the domain $U$
according to normal information diffusion. The normal information diffusion expression is as
shown in Equation (2):

$$f(x) = \frac{1}{h\sqrt{2\pi}} exp\left[-\frac{(x_i - u_j)^2}{2h^2}\right] \quad i = 1,2,...,m; j = 1,2,...,n \tag{2}$$

where $h$ is the information diffusion coefficient, and different values are taken according to
the size of the sample ( $h = 0.8146 \times (b - a), m = 5; h = 0.5690 \times (b - a), m = 6; h =$
$0.4560 \times (b - a), m = 7; h = 0.3860 \times (b - a), m = 8; h = 0.3362 \times (b - a), m = 9; h =$
$0.2986 \times (b - a), m = 10; h = 2.68516 \times (b - a), m \geq 11.$) (Huang,2012).
The domain U obtains the information from the mortality rate sample matrix X with the
normal diffusion. After this, the sample information is normalized via the process of normal
information diffusion. We acquired the discretization information of each domain point $u_j$.
Therefore, the mortality rate expectation $E(S_t)$ can be denoted as follows:

$$E(S_t) = \frac{\sum_{i=1}^m f_i(u_j) \times \left(\sum_{j=1}^n f_i(u_j)\right)^{-1}}{\sum_{j=1}^n \sum_{i=1}^m f_i(u_j) \times \left(\sum_{j=1}^n f_i(u_j)\right)^{-1}} \times u_j \tag{3}$$

$$i = 1, 2, ..., m; j = 1, 2, ..., n; t = 1, 2, ... 8$$

where $u_j$ is the point of the domain, $S_t$ is the order of the basic earthquake emergency
scenario, and the number of scenarios is 8.
The discretized domain under each scenario is averagely divided into six levels according
to the classification of the type of disaster (emergency situation, crisis situation, minor disaster,
moderate disaster, major disaster, catastrophe (Eshghi and Larson, 2008)). Hence, the uncertainty
of the mortality rate can be expressed as the possibility of each level of the mortality rate. The





probability of each level can be denoted as follows:

$$P(u_\alpha < u \le u_\beta) = \sum_{j=\alpha}^{\beta} \frac{\sum_{i=1}^{m} f_i(u_j) \times \left(\sum_{j=\alpha}^{\beta} f_i(u_j)\right)^{-1}}{\sum_{j=\alpha}^{\beta}\sum_{i=1}^{m} f_i(u_j) \times \left(\sum_{j=\alpha}^{\beta} f_i(u_j)\right)^{-1}} \quad 1 < \alpha < \beta < n \qquad (4)$$

where P is the probability of the level (the interval with $u$ is less than $u_\alpha$ and is equal or
greater than $u_\beta$), $\alpha$ is the minimum value of the discrete level point, and $\beta$ is the maximum
value of the discrete level point.

## 5 Mortality rate in each scenario

The collected historical destructive earthquake sample belongs to scenario $S_1$ (Table 2),
which constitutes the mortality rate matrix X={$2.459 \times 10^{-4}$, $2.758 \times 10^{-4}$, $0.757 \times 10^{-4}$, 0, $0.001 \times 10^{-4}$,
$1.886 \times 10^{-4}$, $0.141 \times 10^{-4}$, $0.023 \times 10^{-4}$, 0, 0}. According to the maximum value and minimum
value of the mortality rate in the matrix and the precision requirements, we selected $0.000 \times 10^{-4}$
as the minimum value, $2.950 \times 10^{-4}$ as the maximun value, and $0.050 \times 10^{-4}$ as the interval value.
Therefore, the domain U={0, $0.050 \times 10^{-4}$, $0.100 \times 10^{-4}$, $0.150 \times 10^{-4}$, …, $2.950 \times 10^{-4}$}.
**Table 2. Historical earthquakes on mainland China under scenario S1**

| Time | | Epicentral location | Magnitude | Number | Number | Mortality rate |
|---|---|---|---|---|---|---|
| Year-month-day | Hour-min-second | | | of fatalities | of victims | |
| 1983-11-07 | 05:09:45 | Shandong Heze | 5.9 | 46 | 187000 | $2.459 \times 10^{-4}$ |
| 1989-10-18 | 03:10:40 | Shanxi Datong | 5.8 | 29 | 105140 | $2.758 \times 10^{-4}$ |
| 1989-11-20 | 03:18:42 | Chongqing Jiangbei | 5.2 | 4 | 52800 | $0.757 \times 10^{-4}$ |
| 1992-11-30 | 01:38:00 | Sichuan Shiqu | 5.4 | 0 | 27000 | 0 |
| 1996-09-25 | 03:24:00 | Yunnan Lijiang | 5.7 | 1 | 7690000 | $0.001 \times 10^{-4}$ |
| 2001-05-24 | 21:10:43 | Yunnan Ninglang | 5.8 | 2 | 10605 | $1.886 \times 10^{-4}$ |
| 2008-08-20 | 05:35:00 | Yunnan Yingjiang | 5.0 | 5 | 355395 | $0.141 \times 10^{-4}$ |
| 2010-01-31 | 05:36:00 | Sichuan Suining | 5 | 1 | 437000 | $0.023 \times 10^{-4}$ |
| 2011-11-01 | 00:21:28 | Xinjiang Yining | 5.6 | 0 | 143000 | 0 |
| 2012-12-07 | 22:08:00 | Xinjinag Ruoqiang | 5.1 | 0 | 29751 | 0 |

According to the normal information diffusion (Equation (1)), the information carried by
the mortality rate sample matrix X is spread to the domain U. Thereafter, the sample information



is normalized, and we can accquire the discretization information of each sample. Based on
Equation (2), calculating the probability of each domain by weighting the information points and
the mortality rate expectation, the mortality rate expectation under scenario S1 is 0.839. The
mortality rate expectation of all the scenarios can be acquired using the same process. The sample
size and the mortality rate expectation of each scenario are shown in Table 3.

**Table 3.    Sample size and mortality rate expectation in each scenario**

| Scenario S | S1 | S2 | S3 | S4 | S5 | S6 | S7 | S8 |
|---|---|---|---|---|---|---|---|---|
| Sample size | 10 | 32 | 33 | 50 | 19 | 27 | 5 | 7 |
| Moritaty rate expectation | $8.4 \times 10^{-5}$ | $6.06 \times 10^{-5}$ | $1.44 \times 10^{-5}$ | $0.914 \times 10^{-5}$ | $43.2 \times 10^{-5}$ | $7.95 \times 10^{-5}$ | $300 \times 10^{-5}$ | $100 \times 10^{-5}$ |

**6 Quantification of uncertainty in mortality rate estimation**
The rapid estimation of earthquake fatalities is vital for emergency response during the early
hours following the event. We can know both the actual record for the historical earthquakes as
well as the empirical model-estimated fatalities for the historical events. There is a small
difference among the different empirical models as long as the empirical model can answer
critical questions, such as whether a particular earthquake requires a response, and if so, at what
level (level 1, level 2, level 3, level 4). With the addition of a rapid estimation model based on
scenario analysis, we have also proposed a fatality-based alert scale that provides an estimation
of the likelihood of a range of fatalities caused by an earthquake. The overall dispersion is
associated with the model's prediction for the past earthquakes in that country or region, and then
one uses such a measure for determining the uncertainty associated with the model's future
estimates. The estimation for the probability of each mortality rate range is shown in Figure 7.






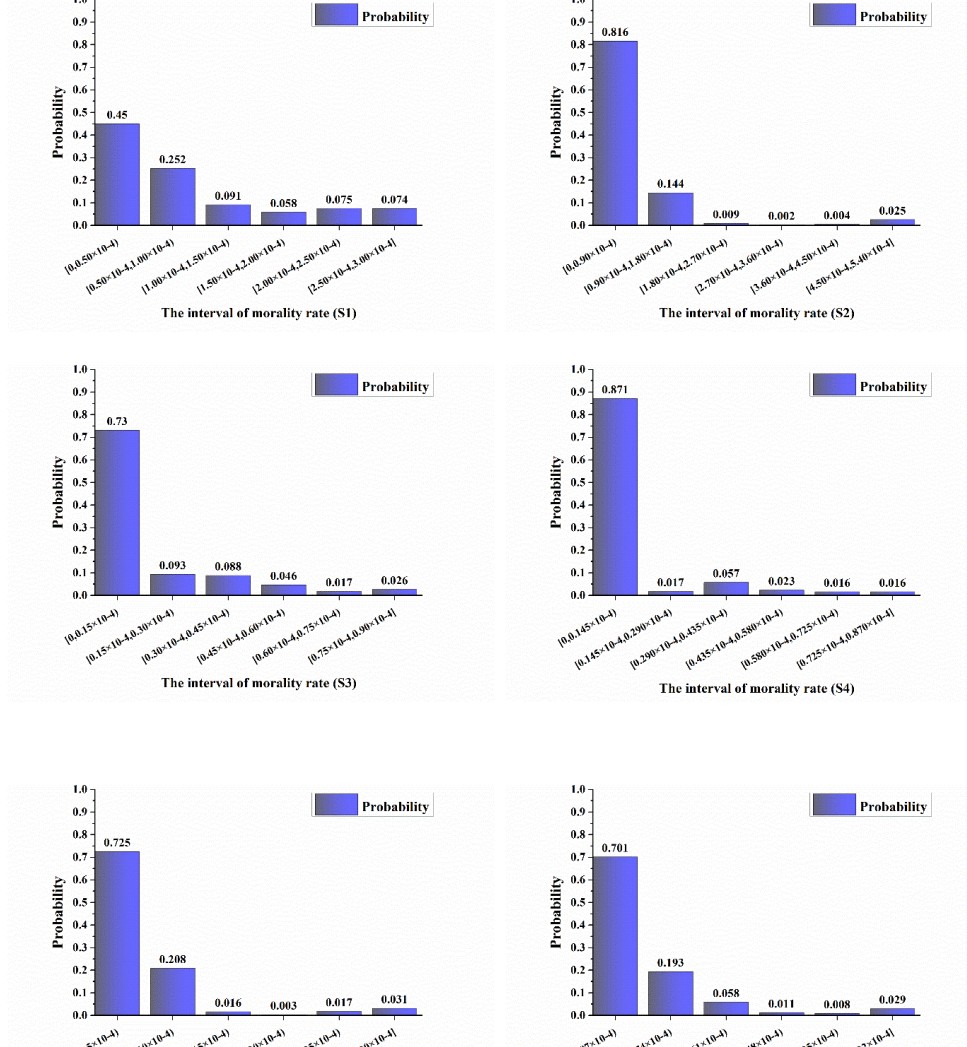



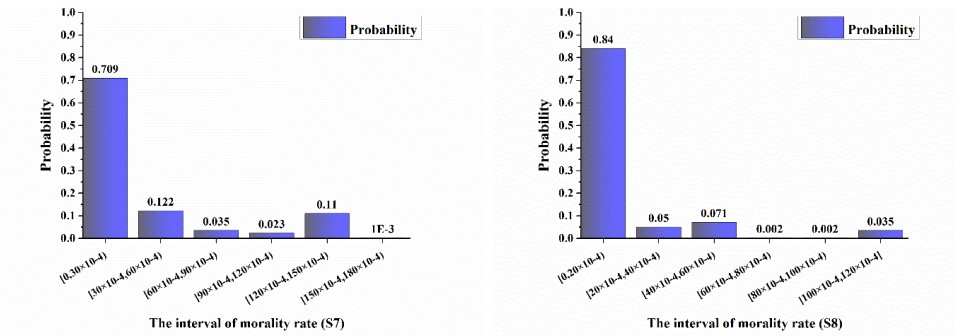

**Figure 7. Probability of the mortality rate under each scenario**

## 7 Verification

The empirical model has been verified using historical earthquakes. Out of a total of 219 earthquakes for which data was collected in this study, 44 (20% of the samples under each scenario were randomly selected) were estimated using the rapid estimation model, and the results are shown in Table 4. Incidentally, we assessed the accuracy of the model via a comparison between the recorded fatalities and estimated fatalities. Among the outliers, the model predicted fewer fatalities for an earthquake (M 6, 9 July 1979) in China, i.e., Jiangsu Liyang, that killed 41 people. At the same time, there were some overestimated fatalities, such as for the earthquake in Hebei Zhangbei (M 6.2, 10 January 1998) and the earthquake in Sichuan Wenchuan (M 8, 12 May 2008). Among the remaining events, the preliminary estimates were within an order of magnitude of the recorded deaths. The number of fatalities calculated using the model was the same order of magnitude as the actual recorded number for more than 95% of the events. The same order of magnitude will not influence the level of the emergency decision, which is very important for rapid post-earthquake rescue.

**Table 4. Verification of historical cases**

| Scenario | Time | Epicentral location | Magnitude | Actual record | Model calculation |
|---|---|---|---|---|---|
| | 2001-05-23 | Yunnan Ninglang | 5.5 | 2 | 1 |
| S1 | 2004-05-04 | Qinghai Delingha | 5.5 | 0 | 0 |
| | 2012-12-07 | Xinjiang Ruoqiang | 5.1 | 0 | 3 |
| S2 | 2012-06-24 | Yunnan Ninglang | 5.7 | 4 | 5 |
| | 1993-08-07 | Sichuan Muchuan | 5 | 0 | 1 |





|  | 2003-10-25 | Gansu Shandan | 5.8 | 9 | 11 |
|---|---|---|---|---|---|
|  | 2007-07-20 | Xinjiang Tekesi | 5.7 | 0 | 5 |
|  | 2011-03-10 | Yunnan Yingjiang | 5.8 | 25 | 12 |
|  | 2013-04-17 | Yunnan Eryuan | 5 | 0 | 7 |
|  | 2013-08-31 | Yunnan Xianggelila | 5.9 | 3 | 5 |
|  | 2006-08-25 | Yunnan Zhaotong | 5 | 1 | 1 |
|  | 2008-12-26 | Yuanan Ruili | 4.9 | 0 | 4 |
|  | 2008-04-21 | Gansu Sunan | 4.2 | 0 | 0 |
| S3 | 2003-11-13 | Gansu Dingxi | 5.1 | 1 | 0 |
|  | 2001-02-23 | Sichuan Yajiang | 5.6 | 3 | 10 |
|  | 2005-08-02 | Yunnan Huize | 5.3 | 0 | 10 |
|  | 1995-03-19 | Xinjiang Heshuo | 5.1 | 0 | 3 |
|  | 1995-04-26 | Sichuan Muchuan | 5.1 | 0 | 0 |
|  | 1996-01-09 | Xinjiang Shawan | 5.6 | 0 | 0 |
|  | 2001-04-12 | Yunnan Shidian | 5.6 | 2 | 1 |
|  | 1996-01-16 | Sichuan Rongchang | 4.3 | 0 | 0 |
|  | 2013-03-29 | Xinjiang Jichang | 5.6 | 0 | 0 |
|  | 1999-11-01 | Shanxi Datong | 5.3 | 0 | 20 |
| S4 | 2011-08-11 | Xinjiang Jiashi | 5.6 | 0 | 0 |
|  | 2012-01-08 | Xinjiang Heshuo | 5 | 0 | 0 |
|  | 1997-01-25 | Yunnan Mengla | 5.1 | 0 | 0 |
|  | 1997-05-31 | Fujian Liancheng | 5.2 | 0 | 2 |
|  | 1995-02-18 | Yunnan Cangyuan | 5.1 | 0 | 0 |
|  | 2013-12-01 | Xinjiang Keping | 5.3 | 0 | 0 |
|  | 2003-05-04 | Xinjiang Jiashi | 5.8 | 1 | 1 |
|  | 1998-01-10 | Hebei Zhangbei | 6.2 | 49 | 116 |
|  | 1989-09-22 | Sichuan Xiaojin | 6.6 | 1 | 23 |
| S5 | 2005-04-08 | Xizang Zhongba | 6.5 | 0 | 2 |
|  | 2015-07-03 | Xinjiang Pishan | 6.4 | 3 | 17 |



| | 2008-10-05 | Xinjiang Wuqia | 6.8 | 0 | 6 |
|---|---|---|---|---|---|
| | 1979-07-09 | Jiangsu Liyang | 6 | 41 | 15 |
| | 1989-04-15 | Sichuan Liangshan | 6.4 | 8 | 3 |
| | 1991-02-25 | Xinjiang Keping | 6 | 0 | 3 |
| S6 | 2003-08-16 | Neimenggu Chifeng | 6.1 | 4 | 33 |
| | 2012-08-12 | Xinjiang Yutian | 6.2 | 0 | 2 |
| | 1995-10-24 | Yunnan Wuding | 6.5 | 58 | 75 |
| S7 | 1976-07-27 | Hebei Tangshan | 7.5 | 242769 | 262540 |
| S8 | 2008-05-12 | Sichuan Wenchuan | 8 | 69227 | 122200 |
| | 2013-04-20 | Sichuan Lushan | 7 | 196 | 254 |

The main purpose of the verification for the uncertainty was to optimize the estimation result.
Furthermore, the possible fatality interval was necessary to provide the basis for emergency
decisions when needing to consider indeterminate factors during the process, particularly when
the main factors for assessment were difficult to acquire. To verify the accuracy of the quantified
results, we used the random selection of 20% of the samples under each scenario. The results
show (Table 5) that under the same scenario, the frequency of events with a small mortality rate
was higher, and the frequency of catastrophic events was lower. There is an advantage of the
model in that the mortality rate distribution can cover all possible historical scenarios. To a certain
extent, this compensates for the lack of extreme events during the fitting of the historical data.
The results were obtained in the form of interval probability statistics, which provide the basis
for the subsequent emergency optimization.
**Table 5. Verification of the probability of the mortality rate interval**

| Scenario | Interval I | Interval Ⅱ | Interval Ⅲ | Interval Ⅳ | Interval Ⅴ | Interval Ⅵ |
|---|---|---|---|---|---|---|
| S1 | 100% | 0% | 0% | 0% | 0% | 0% |
| S2 | 100% | 0% | 0% | 0% | 0% | 0% |
| S3 | 84% | 5% | 11% | 0% | 0% | 0% |
| S4 | 94% | 0% | 3% | 0% | 3% | 0% |
| S5 | 100% | 0% | 0% | 0% | 0% | 0% |





| | | | | | |
|---|---|---|---|---|---|
| S6 | 71% | 29% | 0% | 0% | 0% | 0% |
| S7 | 100% | 0% | 0% | 0% | 0% | 0% |
| S8 | 80% | 0% | 0% | 0% | 20% | 0% |

## 8 Estimation for recent earthquakes

With socio-economic changes, the previous analysis based on historical data may be inconsistent with recent data. Therefore, it is necessary to conduct further verification for the applicability and accuracy of the model using destructive earthquakes that have occurred during recent years. The results of the model calculation were compared to the recorded results. The result and error of the victim estimation is shown in Figure 8. The number of victims calculated via the model is of the same order of magnitude as the recorded number, and the error of the estimation results is less than 30%, which is in line with the requirements of the National Disaster Reduction Committee and the Ministry of Civil Affairs Disaster Reduction Center for the rapid estimation of a disaster.

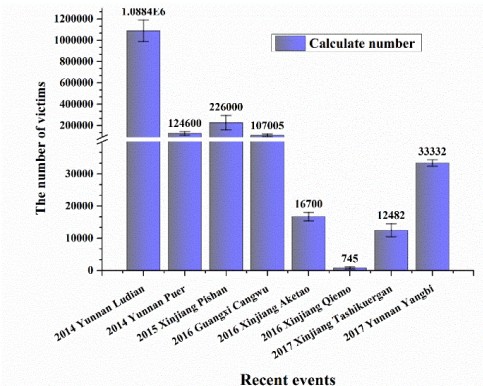

**Figure 8. Estimation of the earthquake victims in recent years**

The number of fatalities during each earthquake was estimated based on the estimation result for the victims. In addition, two models were chosen for comparison, and the selection of the model here considered that the impacts of the empirical models have regionally varied. Thus, we selected two empirical models with Chinese samples, but with different sample numbers and different forms; the comparision results are shown in Table 6. The first method was proposed by Liu et al (2012), which set the epicentral intensity as the main parameter, and the magnitude and average population density were auxiliary parameters in the model. There is a large deviation in



the estimation result of Yunnan Puer (2014). The reason for this may be that the auxiliary
parameter is the average population density in the affected area rather than the unit statistics,
which did not consider the population distribution. The second method was proposed by Xiao
(1991). The overall evaluation result of this eatimation model was good. However, there was a
poor result for Yunnan Ludian (2014). The reason for this was that the sample age chosen by the
model was rather old.The accuracy rate is defined as the total number of events divided by the
number of events for which the estimation results are the same grade as the actual records. The
rapid estimation model based on scenario analysis has a higher accuracy and is more suitable for
rapid estimation via the comparision.

**Table 6.  Estimation results of each method**

| Earthquake events | Victims | | Fatalities | | | |
|---|---|---|---|---|---|---|
| | Actual record | Model calculation | Actual record | Model calculation | First method calculation | Second method calculation |
| Xinjiang Taerkushigan (2017) | 12482 | 14485 | 8 | 1 | 5 | 0 |
| Yunnan Yangbi (2017) | 33332 | 27000 | 0 | 2 | 2 | 0 |
| Guangxi Cangwu (2016) | 107005 | 101778 | 0 | 6 | 14 | 0 |
| Xinjiang Qiemo (2016) | 745 | 1100 | 0 | 0 | 5 | 64 |
| Xinjiang Aketao (2016) | 16700 | 18000 | 0 | 1 | 31 | 0 |
| Xinjian Pishan (2015) | 226000 | 156094 | 3 | 12 | 47 | 1 |
| Yunnan Ludian (2014) | 1088400 | 986439 | 617 | 427 | 1017 | 18 |
| Yunnan Puer (2014) | 124600 | 123000 | 1 | 53 | 471 | 4 |
| Accuracy rate | - | 100% | - | 87.5% | 50% | 75% |

The estimation results of the Yunnan Ludian earthquake (2014) and the Xinjiang
Tashikuergan earthquake (2017) were not the same order of magnitude of the actual records.
These two scenarios should be considered as the extreme events because of their mortality rates.
The fatality interval of Yunnan Ludian (2014) was estimated by the model as [582,680], and the
probability was 0.071. For the Xinjiang Tashikuergan earthquake, the fatality interval was [8,10],
and the probability was 0.026. The interval estimation of the fatalities in the model can consider
the extreme events with larger mortality rates but small probability.




**Table 7. Validation of the model interval**

| Earthquake events | | | Fatalities | | |
|---|---|---|---|---|---|
| Year | Location | Actual record | Model calculation | The interval of fatalities | Probability |
| 2014 | Yunnan Ludian | 617 | 427 | [0,88) | 0.817 |
| 2014 | Yunnan Puer | 1 | 53 | [0,61) | 0.725 |
| 2015 | Xinjian Pishan | 3 | 12 | [0,14) | 0.817 |
| 2016 | Guangxi Cangwu | 0 | 6 | [0,2) | 0.871 |
| 2016 | Xinjiang Aketao | 0 | 1 | [0,9) | 0.725 |
| 2016 | Xinjiang Qiemo | 0 | 0 | [0,1) | 0.871 |
| 2017 | Xinjiang Taerkushigan | 8 | 1 | [0,1) | 0.730 |
| 2017 | Yunnan Yangbi | 0 | 2 | [0,1) | 0.871 |

**9 Conclusion and discussion**
Based on the study of earthquake data from mainland China (1970-2017), we proposed a
new approach for rapidly estimating earthquake fatalities and quantifying the uncertainty. The
main factors of the basic earthquake emergency scenarios were magnitude, intensity (the
relationship between the epicentral intensity and the epicentral fortification intensity) and initial
time, which were used to express the possible earthquake scenarios. For verification of the model,
we not only verified using the recorded number but also presented a comparison to the actual
recorded fatalities of historical earthquakes. The fatality estimation results were mostly of the
same magnitude as the actual record, and the accuracy of the results were higher than that of the
compared empirical model. In addition, the mortality rate interval in the model can effectively
cover the high probability of mortality as well as extreme events. Based on the current study, the
following aspects were mainly improved:
1. During the actual emergency process, the information on on-site earthquakes will be acquired
as time progresses. Therefore, how to update the results with the updated information is in need
of further study.
2. With the development of remote sensing and unmanned aerial vehicle (UAV) technology,
images can be used after the earthquake for damage estimation. The real-time evaluation results





of regional earthquake damage can be acquired. We can obtain relatively accurate information

for local regions. Thus, how to extrapolate the local information to estimate the global demand

may need further study.

Xiaoxue Zhang analyzed and historical data and also guided focus model design and

implementation. Hanping Zhao, Fangping Wang, Zezheng Yan, Sida Cai, Han Wang

& Xiaowen Mei guided focus model design and implementation.

Competing interests. The authors declare they have no conflicts of interest

*Acknowledgements.* This project was supported by the National Natural Science Foundation of

China (NSFC: 41471424) and the National Key Research and Development Program of China

(2017YFB0504102).

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
