# Peer review of "Quantification of uncertainty in rapid estimation of earthquake fatalities"

_Natural Hazards and Earth System Sciences, 2018_

## Referee Comment (RC1) · Anonymous Referee #1 · 17 Aug 2018

Review of
**Quantification of uncertainty in rapid estimation 1 of earthquake fatalities based on scenario analysis
By Xiaoxue Zhang et al.**

Submitted to Natural Hazards and Earth System Sciences (NHESS).
for a Special Issue
*Advances in computational modelling of natural hazards and geohazards*

**General comments**

Clearly, the quantification of uncertainty when rapidly estimating earthquake fatalities is an important goal. There are many important uses and users of rapid earthquake losses assessments. Hence, the topic is of importance and is timely given improved data and computational resources to bring to bear for such calculations. Unfortunately, this study does not further the science of human casualty estimation, nor does it put forth useful modeling tools to address this challenge. The authors inadequately describe the limitations of their own model and those of earlier studies that rely on inadequate relationships (predictor variables) for estimating fatalities. They further ignore significant progress that has been made more recently. In short, the authors rely on shaking computed based on functions of magnitude, time of day, and epicentral intensity rather than employing modern ground motion prediction strategies.

This manuscript is not ready for publication in its current form due to both considerable grammatical issues and scientific limitations. Many sentences require grammatical improvements and many more are at best noninformative due to language, or more general communication issues. Hence, any resubmission needs to go through significant editorial efforts. Yet, as clarified below, the real limitations of this study are with the proposed model, or at least my limited understanding of the model given the descriptions as presented. Below I enumerate some of my concerns.

**Specific comments**

1.  The Abstract does not provide any useful information about the study nor the model presented. I do not consider this manuscript ready for peer review:

    Abstract:
    9 The rapid estimation of earthquake fatalities using earthquake parameters is the core
    10 basis for emergency response. However, there are numerous factors affecting earthquake
    11 fatalities, and it is impossible to obtain an accurate estimation result. The key to solve this
    12 problem is quantifying the uncertainty. In this paper, we proposed a new method to estimate
    13 earthquake fatalities and quantify the uncertainty based on basic earthquake emergency scenarios.
    14 The accuracy of the model is verified by earthquake that occurred during recent year. The
    15 preliminary analysis and comparison results show that the model is more effective and reasonable
    16 and can also provide a theoretical basis for post-earthquake emergency response.

2.  An early, key description of the methodology is particularly opaque to the reader:

    L69. "*The basic scenarios are constructed using the magnitude, the initial time and the relationship between the epicentral intensity and the epicentral fortification intensity, and these scenarios consider combinations of parameters.*" *Epicentral intensity** is an archaic concept, and *fortification intensity* is not defined, nor referenced, and the terminology is unknown to this reviewer (who has been studying earthquake macroseismic intensity for 25 years). Either that term is used in China but not elsewhere and is not explained, or it is a misuse of key terminology (but I cannot tell). "*consider a combination of parameters*" does not inform the reader at all. Even the term "*initial time*" is not explained.

    **epicentral intensity*, normally referred to as $I_o$, e.g., the intensity at the epicenter, does not reflect any of the modern understanding of site amplification, distance to fault, and the fact that the epicenter---particularly for larger earthquakes considered---in this study, does not necessarily represent the worst shaking levels experienced. Standard operating procedures worldwide use ShakeMap (USGS) or equivalent strategies, or at least employ modern ground motion

prediction equations (GMPEs) with some knowledge of site conditions. None of these modern tools are mentioned in this paper not are these basis seismological considerations described or addressed.

3. Or, also early in the manuscript: L92. *A basic scenario combination can better express the relationship between the parameters and earthquake fatalities. Then, information diffusion theory was used to diffuse the sample data based on the basic scenarios considering the temporarily nonmeasurable factors and the extreme event under each scenario."* This reveals very little to this reviewer. It is difficult to sense of either sentence. Please rewrite.

4. Later on, L155: *However, when dividing the samples into each scenario, the sample size will be small, and it is difficult to obtain the relation equation using traditional mathematical statistics. Therefore, the indirect approach of this study consisted of information diffusion theory to obtain the mortality rate.* If traditional statistics can't be used, isn't the problem is ill-posed? Explain why diffusion theory would change this?

5. L30. *At present, the methods for estimating earthquake fatalities mainly include analytical, semi-analytical and empirical models (Federal Emergency Management 30 Agency (FEMA), 2005).* → Quoting an outdated 2005 article for "present" methods does not seem appropriate.

6. L47. *Jaiswal et al. (2009) established a mortality model based on population distribution according to rebuilt earthquake case scenes and studied regional earthquake cases (Jaiswal et al. 2010).* In additional to grammatical problems, this does not adequately explain those Jaiswal et al studies.

7. L50. *Generally speaking, the current empirical model for fatality estimation is derived from available historical data and relies on parameter regression analysis.* This implies that the prior reference to Jaiswal et al (2009, 2010) are parametric regression studies (as *are* the *earlier* studies cited). That is incorrect. Jaiswal's studies are significantly more advanced in terms of hazard input (ShakeMap) and empirical model building than those proposed in the current manuscript, at least from what I can glean from the descriptions herein.

8. L52. *First, it will ignore extreme events when there is lack of historical data. Second, most models consider fewer factors and do not consider the influence between know factors and possible unknown factors.*

9. L58. *During recent years, the study of uncertainty in the estimation of earthquake fatalities has mainly regarded the qualitative.* These statements are incorrect. Two of the references cited (Jaiswal et al 2009, 2010), among others (Wald et al. 2010), specifically address uncertainty in fatalities estimates. Extreme events are fine as long as the model is calibration. For areas without data (lack of damaging earthquakes), of course empirical models are inadequate. This point was not articulated.

10. L62. *There are many linguistic uncertainties when describing the uncertainty in terms of vagueness and context, which can result in an inaccurate qualitative description.* Wald et al (2010) specifically address these issues. This manuscript under review does not address any of them directly.

11. L63. *The numerical quantification of uncertainty is possible for emergency decision making when the information is partial or not quantifiable during the process of estimation.* I don't understand this sentence.

12. L71 *This study not only breaks the traditional empirical model form but also quantifies the uncertainty in the estimation results.* Actually, the form of model used (again, as I try to interpret it) *is* traditional and is no longer used, and uncertainty of fatalities estimates *has* been done in the past (see #9 above). Despite the claim that this has not been done before, it has. Yet, nowhere is the current study *is* uncertainty actually quantified or well described. The only quantification of the model is an "accuracy rate" based on hindcasting a subset of the events for one sampled subset.

13. L78. *number of victims.* "Victims" is not defined. Is it injured (to what degree), displaced persons?

14. L79. *The disaster information was derived from EM-DAT (http://www.emdat.be/), and the earthquake parameters were obtained from PAGER ([https://www.pager.com/](https://www.pager.com/)).* What is meant by *disaster information"?* Which *earthquake parameters* where obtained from PAGER? PAGER does not provide epicentral intensity, so where did that come from? PAGER is NOT [www.pager.**com**](www.pager.com).

15. L86. "*Scholars have discussed the factors that affect earthquake fatalities, which include magnitude, intensity, initial time, population exposure, housing fragility, and individual factors (Oike, 1991; Nichols, 2003)*". These are antiquated references given the rapid evolution of this science.

16. L91. "*Basic earthquake emergency scenarios were constructed based on a combination of the main factors.*" Please rewrite; I can't understand.

17. L218. *We collected data on destructive earthquakes that caused casualties in China from 1970 to 2017.* Wasn't it stated on L.79 that these data were collected by USGS (PAGER)? Or if you did collect such data, what where the sources?

18. L97. *Via qualitative analysis using the collected data, the main factors affecting earthquake fatalities were acquired. There is an approximately linear relationship between the magnitude and the number of fatalities (Figure 2). As the*

*magnitude increases, the number of fatalities increases*. This correlation is known to be inadequate. Magnitude is a poor proxy for shaking for a number of reasons, and fatalities are related to shaking damage, *not* magnitude. Depth, distance to the fault, population exposure at a given shaking intensity and vulnerability are known predictor variables in modern studies.

19. L101. *The relationship between the number of fatalities and the initial time is relatively vague.* Fatalities vs time of day *cannot simply be related* without considering the population exposure per intensity level at the time of the earthquake for each event. Nighttime events *should* be more deadly than during the daytime, but one event at night may be in an area of high population and one during the day in low; one needs to normalize for these other important variables.

20. L108. *Based on the aforementioned analysis, the magnitude, epicentral intensity and initial time were selected as the main parameters used to establish the basic earthquake emergency scenarios*. In the sentences above this, Time of Day was should to not be correlative. Why is it then used? Magnitude should not be used (as described in #18), and there are problems with epicentral intensity (described above). So, it is unlikely that these parameters provide a robust fatality estimate model.

21. L112. *The magnitude was divided into three levels ($4.5 \leq M < 6$, $6 \leq M < 7$ and $7 \leq M \leq 8$ (M means magnitude)) according to the principle of magnitude 112 division in the earthquake emergency programming of China (The National Earthquake Emergency Plan, 2012).* The citation for these terms is "The national earthquake emergency plan" which is in Chinese. Moreover, not only is magnitude not a proper predictor variable, bins of M4.5 to M6.0 or M6 to M7 cover enormous ranges; binning them together is not justified or justifiable. (M6 fault is ~10-15 km in length; M7 is ~60 km in length). Likewise, would one expect shaking (and thus, fatalities, all other things being equal) from an M4.5 to be in the same category as shaking from an M6.0 event?

22. *L115. The relationship between magnitude (M) and epicentral intensity ($I_0$) is as follows : $M = 0.58I_0 + 1.5$ (GB/T17742).* The Reference given is: "*GB/T17742, The Chinese seismic intensity scale, 2008.*" I don't know what this source is, but the Chinese Seismic Intensity Scale has not been published, as far as this reviewer knows. The Reference is also in Chinese. I cannot examine either the scale or equation cited.

23. L132. The term "*basic earthquake emergency scenarios*" is a term not used as far as I know, and it's not clear what exactly it refers to.

24. L139. *We needed a functional form describing the fatalities with the victim and moritality rate*. In addition to grammatical and typos, I cannot decipher this sentence.

25. L143. *seismic intensity elliptical attenuation model.* Elliptical attenuation needs justification; it is not used in any modern ground motion prediction equations.

26. L265. *The overall evaluation result of this eatimation model was good.* This statement does not provide any information to the reader. What evaluation, with what statistics and what does "good" mean?

27. L277. *The interval estimation of the fatalities in the model can consider the extreme events with larger mortality rates but small probability*. Please rewrite. I don't know what this means.

28. Table 5. I don't understand this table. Please explain.

29. Table 6. First "victims" is not defined in the paper. Does this mean displaced? Injured? If either, data for displaced persons or injuries is so uncertain that it challenges credibly that the model can accurately predict them (100% as reported there). If is not believable that one can get an exact answer to an uncertain problem. Please define "victims" and explain how it is possible to calculate this number especially with such a simplified shaking model (elliptical shape)?

30. Table 7. Define "interval of fatalities".

31. Conclusions. L291. *Based on the current study, the following aspects were mainly improved: 1. During the actual emergency process, the information on on-site earthquakes will be acquired as time progresses. Therefore, how to update the results with the updated information is in need of further study. 2. With the development of remote sensing and unmanned aerial vehicle (UAV) technology, images can be used after the earthquake for damage estimation.* These are not "conclusions" since these topics were not even discussed in the paper. Please clarify.

**Technical Corrections**

1. There are numerous errors in the References, including, but not limited to:
   a. Wald D J, Earle P S, Allen T I, et al. Development of the US Geological Survey's PAGER system is given a rather wrong journal title (*Journal of Automated Chemistry*)???
   b. Jaiswal et al is cited incorrectly as "U.s. Geological Survey" (with no report series or journal).
   c. GB/T17742, The Chinese seismic intensity scale, 2008.?
   d. At least seven of the key References are in Chinese.
   e. Larson R C, Eshghi K. is cited as Eshghi and Larson in the manuscript.

2. There is a large number of typos in the manuscript that any word processor should catch. I've not enumerated them but please use a spell-checker.

Review References Cited

Wald, D. J., K. Jaiswal, K. D. Marano, and D. Bausch (2010). An Earthquake Impact Scale, *Natural Hazards Review*, 12, 125-139, doi:10.1061/(ASCE)NH.1527-6996.0000040

---

## Referee Comment (RC2) · Anonymous Referee #2 · 27 Sep 2018

The paper presents a method to estimate earthquake fatality and to quantify the uncertainty in the estimation. The method is statistical analysis of the empirical fatality data. A sophistication is related to the classification of scenarios and further statistical modelling of the fatalities given a scenario – it is a two-stage modelling process. The separation of the data into different scenarios may reduce the scatter of the data within a scenario and thus tend to behave 'better' for statistical characterisation. This is a valid method – however, this itself seems to be insufficient to claim the innovation as a new research – it is more like an incremental research. This comment is based on the deficiency of the hazard modelling aspect of the proposed method and the lack of the demonstration of the robustness/quantitative performance of the method. In addition,

the method is quite local (specific to China), which casts the doubt with regard to its applicability to other seismic regions. From this perspective, the paper is suitable for Chinese journals, not international ones. These comments are elaborated below.

The accurate mortality estimates are important but are not the sole source of critical elements in rapid fatality estimates. The hazard and exposure elements are also important. The proposed method only uses the macroscopic earthquake information (magnitude and source intensity). Modern rapid earthquake impact assessment methods use site-specific estimates of ground shaking, local site conditions, and if available, real-time assimilation of ground motion data and/or human-based intensity observations. The critical differences between the method proposed in this method and these modern methods are the use of local seismic shaking information in estimating the earthquake impact (which is distributed across space). The lack of this element in the proposed method is deficient.

The robustness of the performance of the methods is not well demonstrated. The split of the data into calibration and validation data is fine but this can be more rigorous – for example, a comprehensive cross validation should be carried out and quantify the performance.

The title is misleading because, essentially, the proposed method and its applications are mainly for China, not other parts of the world. Of course the method can be adopted for other parts of the world. However, the paper is insufficient to establish the innovation against existing methods, such as USGS. Methodologically, it is not clear how novel this method is. It appears that there are improvements with regard to the existing methods in China but this claim is not necessarily justified against, for instance, PAGER method and something similar. Ideally, the proposed method should be compared with the state-of-the-art methods (for the same conditions) – it is certainly possible that the proposed method performs better than global ones because the former is calibrated using local data. These comparisons should be done on a fair, objective basis.

[Figure]

It is unclear how the epicentral fortification intensity is defined. Some texts in Section 3 are repetitive and redundant. Figures 2-5 – the vertical axes should be in log10, not natural log so that readers can convert the numbers into the original scale. Figure 6: what is 'oder'? The validity of normal distribution is not clearly explained (in light of data characteristics). Is this a valid assumption? Robustness of the results shown in Table 3 is not well quantified.

―――――――――――――――――

---

## Editor Comment (EC1) · A. J. Kettner (Editor) · 28 Sep 2018

Dear Zhang, co-authors,

The work you describe in "Quantification of uncertainty in rapid estimation of earthquake fatalities based on scenario analysis'" touches upon an important topic, assessing the quality of estimations of earthquake fatalities soon after such an event. To do so the study proposes to use a model that is based on basic earthquake emergency scenarios. However, much progress has been made in recent years and I agree with the reviewers, that the study is lacking to incorporate more modern used rapid earthquake impact assessment methods, that are more refined and enriched by incorporating local

obtained information. The authors might have deliberately chosen not to incorporate these more modern assessment methods but then should have provided a thorough reasoning for not doing so and highlight what makes their presented model more advanced against these modern assessment methods. Without this the study is less innovative and as such doesn't contribute to significant new scientific insights.

Best regards, Albert Kettner Guest editor special issue NHESS

---

## Author Comment (AC1) · 9 Oct 2018

Thank you very much for reading the article carefully and putting forward many key points. In response to your questions and opinions, we make the following responses: 1. For the point 2. ' The epicentral intensity' is defined as intensity of earthquake source in earthquake, which refers to the degree of earthquake damage to buildings such as ground and buildings. 'Fortification intensity' is defined as the seismic precautionary intensity, it must be determined according to the size of the city in which the building is located, the type and height of the building and the planning of the local seismic fortification area. 'The initial time ' is defined as the moment of

earthquake is the moment when the source body starts to crack.  2.For the point 4. The information diffusion theory is a domain-specific information distribution theory for small sample problems.  It can expand the sample and obtain the probability of occurrence of a particular domain. The specific theory is shown in the formula (1-3) in this paper. 3. For the point 7. We want to express our approach is different from the traditional empirical approach, not based on parameter fitting, not to negate previous studies. 4. For the point 13. 'Victim' means the people affected by the disaster in the specific area. 5. For the point 14. The epicentral intensity is not come from PAGER, the paper sated in line 117, the epicentral intensity is calculated by the empirical formula. 6. For the point 5 and 15. The selected references are long-standing, but the research results are widely circulated, and the research results are affirmed by the industry. 7. For the point 17. The data of typical earthquake cases selected for verification are mainly from the book named "Assessment Compilation of Earthquake Disaster Losses in Mainland China", which compiled by the Earthquake Disaster Emergency Rescue Department of China Earthquake Administration. 8.For the point 18. In this paper, we use the first-time acquired basic seismic parameters to evaluate the earthquake as it occurs before other loss data are obtained, so we selected the parameters, which are easy obtain.  Magnitude is the easiest way to get.  9.  For point19.20.21. In this paper, we use the first-time acquired basic seismic parameters to evaluate the earthquake as it occurs before other loss data are obtained, so we select the intensity rather than the ground motion.  Intensity is more macroscopic. And the relationship between the epicentral intensity and the fortification intensity can indirectly express the building losses. 10.For point 20. There is no correlation between the focal depth and earthquake fatality. The fitting results in this respect will be added to explain in the paper. 11.For the point 22. This is an empirical relationship between the epicenter intensity and magnitude of earthquakes in China, as described in the paper. 12. For the point 23. The term "basic earthquake emergency scenarios" base on scenario analysis and data used in the article, all possible scenarios are combined with parameters. 13.For the point 26. In this paper, the validated assessment results

are good in the same order of magnitude without affecting the start-up of emergency plans in China. 14. For the point 27. In this paper, the validated evaluation results are in the same order of magnitude, and do not affect the proportion of the evaluation results in different regions of China's emergency preparedness initiation. 15. We will modify the errors that you put forward in the manuscript.

Please also note the supplement to this comment:
https://www.nat-hazards-earth-syst-sci-discuss.net/nhess-2018-187/nhess-2018-187-AC1-supplement.pdf

---

## Author Comment (AC2) · 9 Oct 2018

Thank you very much for reading the article carefully and putting forward many key points. In response to your questions and opinions, we make the following responses: 1.This method is not purely for China, because the model data is used in China, so the results of some parameters are for China. But if we use other countries' data, we can adjust the parameters according to the method. And the difference between earthquake casualties is very large, so the different models are to consider the difference between time and space. 2. In this paper, we use the first-time acquired basic seismic parameters to evaluate the earthquake as it occurs before other loss

data are obtained, so we select the intensity rather than the ground motion. Intensity is more macroscopic. 3. In this paper, two widely used empirical evaluation models in China are selected for comparison. The principle for the model selection is according to the data, because the empirical model has a great relationship with the selected data, so the model with the data in the same region was selected. 4. We will modify the errors that you put forward in the manuscript.

Please also note the supplement to this comment:
https://www.nat-hazards-earth-syst-sci-discuss.net/nhess-2018-187/nhess-2018-187-AC2-supplement.pdf

---

## Author Comment (AC3) · 16 Oct 2018

General comments This manuscript is not ready for publication in its current form due to both considerable grammatical issues and scientific limitations Response: This method is not purely for China, because the model data is used in China, so the results of some parameters are for China. But if we use other countries' data, we can adjust the parameters according to the method. And the difference between earthquake casualties is very large, so the different models are to consider the difference between time and space. And Grammar, punctuation and formulation issues have been improved. Specific comments 1.The Abstract does not provide any useful

information about the study nor the model presented. Response: In the abstract, we stated the process of building the model and the superiority of the model in the result validation. 2. An early, key description of the methodology is particularly opaque to the reader: Response: I had made amendments in the manuscript text, please see in  section, line 70-77, page 3. 3. Or, also early in the manuscript: L92. A basic scenario combination can better express the relationship between the parameters and earthquake fatalities. Then, information diffusion theory was used to diffuse the sample data based on the basic scenarios considering the temporarily no measurable factors and the extreme event under each scenario." This reveals very little to this reviewer. It is difficult to sense of either sentence. Please rewrite. Response: I had made amendments in the manuscript text, please see in < Basic earthquake emergency scenarios > section, line 96-101, page 4. 4. Later on, L155: However, when dividing the samples into each scenario, the sample size will be small, and it is difficult to obtain the relation equation using traditional mathematical statistics. Therefore, the indirect approach of this study consisted of information diffusion theory to obtain the mortality rate. If traditional statistics can't be used, isn't the problem is ill-posed? Explain why diffusion theory would change this? Response: The information diffusion theory is a domain-specific information distribution theory for small sample problems. It can expand the sample and obtain the probability of occurrence of a particular domain. The specific theory is shown in the formula (1-3) in this paper. 5. L30. At present, the methods for estimating earthquake fatalities mainly include analytical, semi-analytical and empirical models (Federal Emergency Management 30 Agency (FEMA), 2005). à Quoting an outdated 2005article for "present" methods do not seem appropriate. Response: I had made amendments in the manuscript text, please see in  section, line 30-34, page 2. 6. L47. Jaiswal et al. (2009) established a mortality model based on population distribution according to rebuilt earthquake case scenes and studied regional earthquake cases (Jaiswal et al. 2010). In additional to grammatical problems, this does not adequately explain those Jaiswal et al studies. Response: I had made amendments in the manuscript text, please

see in  section, line 40-49, page 2. 7. L50. Generally speaking, the current empirical model for fatality estimation is derived from available historica ldata and relies on parameter regression analysis. This implies that the prior reference to Jaiswal et al (2009,2010) are parametric regression studies (as are the earlier studies cited). That is incorrect. Jaiswal's studies are significantly more advanced in terms of hazard input (ShakeMap) and empirical model building than those proposed in the current manuscript, at least from what I can glean from the descriptions herein. Response: We want to express our approach is different from the traditional empirical approach, not based on parameter fitting, not to negate previous studies. I had made amendments in the manuscript text, please see in  section, line 40-49, page 2. 8. L52. First, it will ignore extreme events when there is lack of historical data. Second, most models consider fewer factors and do not consider the influence between know factors and possible unknown factors. Response: Here we want to express the advantage of our model, and what limitation of the most empirical model. 9. L58. During recent years, the study of uncertainty in the estimation of earthquake fatalities has mainly regarded the qualitative. These statements are incorrect. Two of the references cited (Jaiswal et al 2009, 2010), among others (Wald et al. 2010), specifically address uncertainty in fatalities estimates. Extreme events are fine as long as the model is calibration. For areas without data (lack of damaging earthquakes), of course empirical models are inadequate. This point was not articulated. Response: I had made amendments in the manuscript text, please see in  section, line 40-49, page 2. 10. L62. There are many linguistic uncertainties when describing the uncertainty in terms of vagueness and context, which can result in an inaccurate qualitative description. Wald et al (2010) specifically address these issues. This manuscript under review does not address any of them directly. Response: I had made amendments in the manuscript text, please see in  section, page 2. 11. L63. The numerical quantification of uncertainty is possible for emergency decision making when the information is partial or not quantifiable during the process of estimation. I don't understand this sentence. Response: I had made amendments

in the manuscript text, please see in  section, line 62-64, page 3. 12. L71 This study not only breaks the traditional empirical model form but also quantifies the uncertainty in the estimation results. Actually, the form of model used (again, as I try to interpret it) is traditional and is no longer used, and uncertainty of fatalities estimates has been done in the past (see #9 above). Despite the claim that this has not been done before, it has. Yet, nowhere is the current study being uncertainty actually quantified or well described. The only quantification of the model is an "accuracy rate" based on hindcasting a subset of the events for one sampled subset. Response: I had made amendments in the manuscript text, please see in  section, line 70-71, page 3. 13. L78. number of victims. "Victims" is not defined. Is it injured (to what degree), displaced persons? Response: 'Victim' means the people affected by the disaster in the specific area. 14. L79. The disaster information was derived from EM-DAT (http://www.emdat.be/), and the earthquake parameters were obtained from PAGER (https://www.pager.com/). What is meant by disaster information"? Which earthquake parameters where obtained from PAGER? PAGER does not provide epicentral intensity, so where did that come from? PAGER is NOT www.pager.com. Response: The epicentral intensity is not come from PAGER, the paper sated in line 117, the epicentral intensity is calculated by the empirical formula. And the epicentral intensity was modified as the intensity of epicenter, and the error issues had made amendments in the manuscript text, please see in < Earthquake fatalities in mainland China> section, line 78-80, page 3. 15. L86. "Scholars have discussed the factors that affect earthquake fatalities, which include magnitude, intensity, initial time, population exposure, housing fragility, and individual factors (Oike, 1991; Nichols, 2003)". These are antiquated references given the rapid evolution of this science. Response: The selected references are long-standing, but the research results are widely circulated, and the research results are affirmed by the industry. 16. L91. "Basic earthquake emergency scenarios were constructed based on a combination of the main factors." Please rewrite; I can't understand. Response: I had made amendments in the manuscript text, please see in < Basic earthquake

emergency scenarios > section, line 103, page 5. 17. L218. We collected data on destructive earthquakes that caused casualties in China from 1970 to 2017. Wasn't it stated on L.79 that these data were collected by USGS (PAGER)? Or if you did collect such data, what where the sources? Response: They are the same, but there are some data of typical earthquake cases selected for verification are mainly from the book named "Assessment Compilation of Earthquake Disaster Losses in Mainland China", which compiled by the Earthquake Disaster Emergency Rescue Department of China Earthquake Administration. 18. L97. Via qualitative analysis using the collected data, the main factors affecting earthquake fatalities were acquired. There is an approximately linear relationship between the magnitude and the number of fatalities (Figure 2). As the magnitude increases, the number of fatalities increases. This correlation is known to be inadequate. Magnitude is a poor proxy for shaking for a number of reasons, and fatalities are related to shaking damage, not magnitude. Depth, distance to the fault, population exposure at a given shaking intensity and vulnerability are known predictor variables in modern studies. Response: In this paper, we use the first-time acquired basic seismic parameters to evaluate the earthquake as it occurs before other loss data are obtained, so we selected the parameters, which are easy obtain. Magnitude is the easiest way to get. 19. L101. The relationship between the number of fatalities and the initial time is relatively vague. Fatalities vs time of day cannot simply be related without considering the population exposure per intensity level at the time of the earthquake for each event. Nighttime events should be more deadly than during the daytime, but one event at night may be in an area of high population and one during the day in low; one needs to normalize for these other important variables. Response: In this paper, we use the first-time acquired basic seismic parameters to evaluate the earthquake as it occurs before other loss data are obtained, so we select the intensity rather than the ground motion. Intensity is more macroscopic. And the relationship between the intensity at epicenter and the seismic precautionary intensity can indirectly express the building losses. 20. L108. Based on the aforementioned analysis, the magnitude, epicentral intensity and initial time were

selected as the main parameters used to establish the basic earthquake emergency scenarios. In the sentences above this, Time of Day was should to not be correlative. Why is it then used? Magnitude should not be used (as described in #18), and there are problems with epicentral intensity (described above). So, it is unlikely that these parameters provide a robust fatality estimate model. Response: With the data fitting, there is no correlation between the focal depth and earthquake fatality obviously. The fitting result is not suitable for modeling parameters. 21. L112. The magnitude was divided into three levels ($4.5 \le M < 6$, $6 \le M < 7$ and $7 \le M \le 8$ (M means magnitude)) according to the principle of magnitude 112 division in the earthquake emergency programming of China (The National Earthquake Emergency Plan, 2012). The citation for these terms is "The national earthquake emergency plan" which is in Chinese. Moreover, not only is magnitude not a proper predictor variable, bins of M4.5 to M6.0 or M6 toM7 cover enormous ranges; binning them together is not justified or justifiable. (M6 fault is ~10-15 km in length; M7 is ~60 km in length). Likewise, would one expect shaking (and thus, fatalities, all other things being equal) from an M4.5 to be in the same category as shaking from an M6.0 event? Response: In this paper, we use the first-time acquired basic seismic parameters to evaluate the earthquake as it occurs before other loss data are obtained, so we select the intensity rather than the ground motion. Intensity is more macroscopic. And the magnitude is more easy accept. All the setting is for the rapid assessment without the other loss assessment. 22. L115. The relationship between magnitude (M) and epicentral intensity (I0) is as follows: M = 0.58I0+1.5 (GB/T17742). The Reference given is: "GB/T17742, The Chinese seismic intensity scale, 2008." I don't know what this source is, but the Chinese Seismic Intensity Scale has not been published, as far as this reviewer knows. The Reference is also in Chinese. I cannot examine either the scale or equation cited. Response: This is an empirical relationship between the epicenter intensity and magnitude of earthquakes in China, as described in the paper. 23. L132. The term "basic earthquake emergency scenarios" is a term not used as far as I know, and it's not clear what exactly it refers to. Response: The term "basic earthquake emergency scenarios"

base on scenario analysis and data used in the article, all possible scenarios are combined with parameters. 24. L139. We needed a functional form describing the fatalities with the victim and mortality rate. In addition to grammatical and typos, I cannot decipher this sentence. Response: I had made amendments in the manuscript text, please see in < Basic earthquake emergency scenarios > section, line 134, page 7. 25. L143. seismic intensity elliptical attenuation model. Elliptical attenuation needs justification; it is not used in any modern ground motion prediction equations. Response: In this paper the seismic intensity elliptical attenuation model is uses to estimate the area of the disaster and calculate the population affected. It is not the main factor in the model. 26. L265. The overall evaluation result of this estimation model was good. This statement does not provide any information to the reader. What evaluation, with what statistics and what does "good" mean? Response: In this paper, the validated assessment results are good in the same order of magnitude without affecting the start-up of emergency plans in China. 27. L277. The interval estimation of the fatalities in the model can consider the extreme events with larger mortality rates but small probability. Please rewrite. I don't know what this means. Response: In this paper, the validated evaluation results are in the same order of magnitude, and do not affect the proportion of the evaluation results in different regions of China's emergency preparedness initiation. 28. Table 5. I don't understand this table. Please explain. Response: To verify the accuracy of the quantified results of the uncertainty, we used the random selection of 20% of the samples under each scenario. And the table 5 is the accuracy of each fatality level. 29. Table 6. First "victims" is not defined in the paper. Does this mean displaced? Injured? If either, data for displaced persons or injuries is so uncertain that it challenges credibly that the model can accurately predict them (100% as reported there). If is not believable that one can get an exact answer to an uncertain problem. Please define "victims" and explain how it is possible to calculate this number especially with such a simplified shaking model (elliptical shape)? Response: I had made amendments in the manuscript text, please see in < Basic earthquake emergency scenarios > section, line 77, page 3. 30. Table 7. Define

"interval of fatalities". Response: Interval value of estimation result of the earthquake fatality. 31. Conclusions. L291. Based on the current study, the following aspects were mainly improved: 1. During the actual emergency process, the information on on-site earthquakes will be acquired as time progresses. Therefore, how to update the results with the updated information is in need of further study. 2. With the development of remote sensing and unmanned aerial vehicle (UAV) technology, images can be used after the earthquake for damage estimation. These are not "conclusions" since these topics were not even discussed in the paper. Please clarify. Response: I had made amendments in the manuscript text, please see in < Conclusions > section, line 298, page 17. Technical Corrections Response: I had made amendments in the manuscript text, please see in < Conclusions > section, line 314-367, page 18-20.

Please also note the supplement to this comment:
https://www.nat-hazards-earth-syst-sci-discuss.net/nhess-2018-187/nhess-2018-187-AC3-supplement.pdf

---

## Author Comment (AC4) · 16 Oct 2018

1.In addition, the method is quite local (specific to China), which casts the doubt with regard to its applicability to other seismic regions. From this perspective, the paper is suitable for Chinese journals, not international ones. These comments are elaborated below. The title is misleading because, essentially, the proposed method and its applications are mainly for China, not other parts of the world. Of course the method can be adopted for other parts of the world. Response: This method is not purely for China, because the model data is used in China, so the results of some parameters are for China. But if we use other countries' data, we can adjust the parameters according

to the method. And the difference between earthquake casualties is very large, so the different models are to consider the difference between time and space. 2.The hazard and exposure elements are also important. The proposed method only uses the macroscopic earthquake information (magnitude and source intensity). Modern rapid earthquake impact assessment methods use site-specific estimates of ground shaking, local site conditions, and if available, real-time assimilation of ground motion data and/or human-based intensity observations. Response: In this paper, we use the first-time acquired basic seismic parameters to evaluate the earthquake as it occurs before other loss data are obtained, so we select the intensity rather than the ground motion. Intensity is more macroscopic. 3.This comment is based on the deficiency of the hazard modelling aspect of the proposed method and the lack of the demonstration of the robustness/quantitative performance of the method. Response: In this paper, two widely used empirical evaluation models in China are selected for comparison. The principle for the model selection is according to the data, because the empirical model has a great relationship with the selected data, so the model with the data in the same region was selected. 4. Figure 6: what is 'oder'? Response: I had made amendments in the manuscript text, please see in < Methodology > section, line 274, page 16.

Please also note the supplement to this comment:
https://www.nat-hazards-earth-syst-sci-discuss.net/nhess-2018-187/nhess-2018-187-AC4-supplement.pdf

---

## Author Comment (AC5) · 16 Oct 2018

Quantification of uncertainty in rapid estimation of earthquake fatalities based on scenario analysis Nat. Hazards Earth Syst. Sci. Comments to the Author A. J. Kettner (Editor) The study is lacking to incorporate more modern used rapid earthquake impact assessment methods, that are more refined and enriched by incorporating local obtained information. The authors might have deliberately chosen not to incorporate these more modern assessment methods but then should have provided a thorough reasoning for not doing so and highlight what makes their presented model more advanced against these modern assessment methods. Response: This method is

not purely for China, because the model data is used in China, so the results of some parameters are for China. But if we use other countries' data, we can adjust the parameters according to the method. And the difference between earthquake casualties is very large, so the different models are to consider the difference between time and space. In this paper, we use the first-time acquired basic seismic parameters to evaluate the earthquake as it occurs before other loss data are obtained, so we select the intensity rather than the ground motion. Intensity is more macroscopic. This study built the rapid assessment model based on scenario analysis and quantified the uncertainty in the estimation results.

Please also note the supplement to this comment:
https://www.nat-hazards-earth-syst-sci-discuss.net/nhess-2018-187/nhess-2018-187-AC5-supplement.pdf

---

## Author Comment (AC6) · 16 Oct 2018

Dear editor and two anonymous reviewers: Many thanks for your careful review and constructive suggestions on our manuscript nhess-2018-187 entitled "Quantification of uncertainty in rapid estimation of earthquake fatalities based on scenario analysis " We have taken all your comments into account and responded positively to qualify our manuscript for a potential publication in the journal. Additionally, we have also revised carefully to follow the instructions for authors. All revisions are marked with red in the new manuscript. Our detailed responses are followings in the response letter for red, including three parts . We highly appreciate all your valuable and insightful

comments on the manuscript, and hope that this edition will be more approaching to the journal's requirement. Âă Âă With the best regards! Sincerely yours, Hanping Zhao

Please also note the supplement to this comment:
https://www.nat-hazards-earth-syst-sci-discuss.net/nhess-2018-187/nhess-2018-187-AC6-supplement.zip